# Impact of Neodymium and Scandium Ionic Radii on Sorption Dynamics of Amberlite IR120 and AB-17-8 Remote Interaction

**DOI:** 10.3390/ma14185402

**Published:** 2021-09-18

**Authors:** Talkybek Jumadilov, Bakytgul Totkhuskyzy, Zamira Malimbayeva, Ruslan Kondaurov, Aldan Imangazy, Khuangul Khimersen, Juozas Grazulevicius

**Affiliations:** 1Laboratory of Synthesis and Physicochemistry of Polymers, JSC Institute of Chemical Sciences after A.B. Bekturov, Sh. Valikhanov St. 106, Almaty 050010, Kazakhstan; r-kondaurov@mail.ru (R.K.); imangazy.aldan@mail.ru (A.I.); 2Department of Chemistry, Institute of Natural Science, Kazakh National Women’s Teacher Training University, Aiteke Bi Str. 99, Almaty 050000, Kazakhstan; bakytgul.sakenova@mail.ru (B.T.); malimbayeva.zamira@gmail.com (Z.M.); 3Institute of Natural Sciences and Geography, Abai Kazakh National Pedagogical University, Dostyk Ave. 13, Almaty 050010, Kazakhstan; huana88@mail.ru; 4Department of Polymer Chemistry and Technology, Kaunas University of Technology, K. Donelaičio St. 73, LT119505811 Kaunas, Lithuania; juozas.grazulevicius@ktu.lt

**Keywords:** interpolymer system, industrial ion-exchangers, Amberlite IR120, AB-17-8, remote interaction, mutual activation, sorption, neodymium ions, scandium ions

## Abstract

The aim of the work is to provide a comparative study of influence of ionic radii of neodymium and scandium ions on their sorption process from corresponding sulfates by individual ion exchangers Amberlite IR120, AB-17-8 and interpolymer system Amberlite IR120-AB-17-8. Experiments were carried out by using the following physicochemical methods of analysis: conductometry, pH-metry, colorimetry, and atomic-emission spectroscopy. Ion exchangers in the interpolymer system undergo remote interactions with a further transition into highly ionized state. There is the formation of optimal conformation in the structure of the initial ion exchangers. A significant increase of ionization of the ion-exchange resins occurs at molar ratio of Amberlite IR120:AB-17-8 = 5:1. A significant increase of sorption properties is observed at this ratio due to the mutual activation of ion exchangers. The average growth of sorption properties in interpolymer system Amberlite IR120:AB-17-8 = 5:1 is over 90% comparatively to Amberlite IR120 and almost 170% comparatively to AB-17-8 for neodymium ions sorption; for scandium ions sorption the growth is over 65% comparatively to Amberlite IR120 and almost 90% comparatively to AB-17-8. A possible reason for higher sorption of neodymium ions in comparison with scandium ions is maximum conformity of globes of internode links of Amberlite IR120 and AB-17-8 after activation to sizes of neodymium sulfate in an aqueous medium.

## 1. Introduction

The current high demand for rare-earth metals (REM) has resulted in a significant increase on the initial cost of these metals. REM have various applications in chemical industry, aerospace industry, nuclear engineering, microelectronics etc. [1,2,3]. Neodymium and scandium can be named as one of the most demanded rare-earth metals [4].

Neodymium is a rare earth metal with a silvery-white color with a golden hue, widely used in the manufacture of lasers, glass coloring (toning), and also as a dielectric, demonstrating basically the same characteristics as other elements of the lanthanide group. It is often used as a component of alloys with aluminum and magnesium in rocketry. Magnets made from alloys based on rare earth metals with the chemical composition Nd_2_Fe_14_B have high magnetic properties [5]. In addition, neodymium is used for alloying of special structural alloys and steels, production of dielectric materials, production of thermoelectric materials [6]. Scandium is a silvery-white rare-earth metal [5]. It can be found in such minerals (in trace amounts) as thortveitite (contains over 45% of Sc in oxide form), euxenite and gadolinite in Scandinavia [7] and Madagascar [8]. Scandium resources are found in Australia, Canada, China, Kazakhstan, Madagascar, Norway, the Philippines, Russia, Ukraine and USA [9]. Scandium has no affinity for common ore-forming anions; therefore, it is widely dispersed in the lithosphere and forms solid solutions with low concentrations in more than 100 minerals. The main use of scandium in aluminum-scandium alloys as minor components of the aerospace industry with a scandium content of 0.1% to 0.5% scandium. Scandium is also used for the production of multilayer X-ray mirrors (compositions: scandium–tungsten, scandium–chromium, scandium–molybdenum). Scandium telluride is a very promising material for the production of thermoelements (high thermo-emf, 255 μV/K and low density and high strength). In recent years, refractory alloys (intermetallic compounds) of scandium with rhenium (melting temperature up to 2575 °C), ruthenium (melting temperature up to 1840 °C), iron (melting temperature up to 1600 °C) have acquired considerable interest for aerospace and nuclear technology [10].

The onic radius of Nd^3+^ ions (coordination number VI) is 0.983Å; The onic radius of Sc^3+^ ions (coordination number VI) is 0.745Å [11]. Nowadays, industrial ion-exchange resins are widespread and commercially available [12]. Ion exchangers are solid polymers, insoluble, and swell to a limited extent in electrolyte solutions and organic solvents [13,14]. These high-molecular synthetic compounds with a three-dimensional gel and macroporous structure, which contain functional groups of an acidic or basic nature, capable of ion exchange reactions. Modern sorption technologies for extraction of rare-earth metals in hydrometallurgy are commonly based on application of ion-exchangers [15,16,17,18,19,20,21,22]. As objects of the study widely used industrial ion-exchange resins Amberlite IR120 (cation exchanger) and AB-17-8 (anion exchanger) were chosen as macromolecular sorbents [23].

Amberlite IR120 is a strongly acidic gel-type cation exchange resin based on sulfonated polystyrene. It is used both for softening water (sodium form) and for water demineralization (hydrogen form) in installations with parallel-flow regeneration. Its main characteristics are excellent physical and chemical stability and heat resistance, good ion exchange kinetics and high exchange capacity [24,25,26,27,28,29].

Anionite AB-17-8 is a strongly basic ion-exchange resin with a gel structure. It is used in softening technology and in water demineralization. Differs in good osmotic stability, high chemical resistance to alkalis, acids, oxidizing agents, insoluble in water and organic solvents [30,31,32,33,34].

The complex chemical composition of hydrometallurgical industrial solutions is the most serious problem limiting the metal extraction efficiency. Due to tight crosslinking and strict complementarity to certain ions in the structure of industrial ion-exchangers, the extraction of several valuable components from solutions with sufficient efficiency is impossible. Nowadays, the application of industrial ion-exchangers in hydrometallurgy faced two main issues in terms of the sorption of REM ions [35]:1.The absence of selectivity and universality—sorption of each rare-earth metal ion requires a certain ion exchanger;2.Complicated process of regeneration—each ion exchanger has to be regenerated by continuous washing and renewing its exchange capacity.

Until now, the problem of separation and concentration of rare-earth metals in hydrometallurgy was successfully solved by using both sorption and extraction methods [36]. Among these two methods, the sorption methods are currently preferred due to some advantages over extraction: sorption methods are more environmentally friendly, have a small number of technological cycles in comparison with extraction technologies [37].

At present, different macromolecular sorbents are developed and used in sorption technologies [38] wherein a significant portion of them is focused on selective extraction and separation of different nature metals ions (rare, rare-earth, noble) [39]. The growing global demand for these metals underlines the relevance and practical importance of this study. As it is known, the industrial application of widely used ion-exchange resins is extremely limited, which is associated with the low efficiency and selectivity of synthetic ion exchangers in technological solutions with different ionic compositions [40]. When developing the latest ion exchangers, it is impossible to take into account the ionic composition of industrial solutions. In other words, the main disadvantage of an ion exchanger is that it can be used to remove only one specific metal from a solution. Earlier [41], a new phenomenon of the influence of two rare-crosslinked polymer hydrogels of different nature, placed in a common aqueous medium and separated by a microporous filter (excluding their direct contact) was discovered and called the “long-range effect” (remote interaction phenomenon) [42,43,44,45].

In this regard, the aim of this work is study of the influence of ionic radius of the rare-earth metals on the effect of remote interaction of two commercially available ion exchangers during sorption of rare-earth elements (on the example of neodymium and scandium).

As a result of extensive studies of the rare-earth metals sorption by interpolymer systems consisting of crosslinked polyelectrolytes that undergo a remote interaction, the following advantages of new sorption methods were established [46,47]:(1)Possibility of development of selective sorbents (or sorption systems) for selective sorption of aimed metal ion;(2)Possibility of developing several selective sorbents for sorption of different metals ions on the basis of one interpolymer system;(3)Possibility of application of rather cheap industrial ion-exchangers for creation of interpolymer systems for further sorption of aimed metals ions;(4)Remote interaction provides the transition of each initial component into highly ionized state in interpolymer systems. Each ionized (activated) ion-exchanger can be used as an independent sorbent, which have high sorption properties.(5)Activated ion-exchangers further can be used for creation of new interpolymer systems selective to other ions.

The mentioned above advantages of remote interaction effect over the existing sorption methods can be the basis for development of highly effective sorption methods for selective technology for the extraction of target metals ions from industrial hydrometallurgical solutions.

## 2. Materials and Methods

### 2.1. Materials

#### 2.1.1. Ion Exchangers

Chemical structures of the cation exchanger Amberlite IR120 is presented on Figure 1:

The Amberlite IR120 ion-exchanger was synthetized by the Sigma-Aldrich company (Saint-Louis, MO, USA).

The chemical structure of the anion exchanger AB-17-8 is presented on Figure 2:

The AB-17-8 ion-exchanger was synthetized by LLP Laborfarma (Almaty, the Republic of Kazakhstan).

#### 2.1.2. Preparation of Interpolymer System

The scientific basis of the creation of the interpolymer systems is a phenomenon of the remote interactions of macromolecules. In the study, the interpolymer system, Amberlite IR120-AB-17-8, is based on the initial cation and anion exchangers. The preparation process of the interpolymer system as follows (in stages):(1)The dispersion of each ion-exchange resin undergoes crushing in the analytic mill (separately from each other) to the particles that are larger than 120 µm and smaller than 180 µm.(2)The obtained dispersions of the macromolecular structures are put into specially made polypropylene cells (filters) with pores of 100 µm.(3)The mentioned cells with Amberlite IR120 and AB-17-8 dispersions are put as drawn on Figure 3—in distance of 2 cm in front of each other.

The important moment is when the total amount of the dispersion (for both cases—presence of individual ion-exchanger or presence of interpolymer system in the corresponding salt solution) is constant and equals 6 mol. Analysis of the obtained data will be more convenient if the ratios of ion-exchange resins in the interpolymer system are considered as molar concentrations.

#### 2.1.3. Calculation of Sorption Parameters

Model salts solutions–neodymium sulfate hydrate and scandium sulfate hydrate (C = 100 mg/L in both cases) were used for sorption experiments. Solutions were prepared with application of deionized water (χ = 10 µS/cm; pH = 6.98).

The mentioned rare-earth metals ions extraction degree was calculated in accordance with the Equation:$$\eta =\frac{C_{0}-C_{e}}{C_{0}}*100\%$$
where C_0_–initial concentration of Nd^3+^ or Sc^3+^ ions, mg/L; C*_e_*–equilibrium concentration of Nd^3+^ or Sc^3+^ ions, mg/L.

The total polymer chain binding degree of Amberlite IR120, AB-17-8 and interpolymer system based on these ion-exchange structures was calculated in accordance with the equation:$$\theta =\frac{\nu_{sorbed}}{n_{1}\nu_{2}+n_{2}\nu_{2}}*100\%$$
where *ν_sorbed_*—amount of the sorbed Nd^3+^ or Sc^3+^ ions, mole; *ν*_1_—quantity of Amberlite IR120, mol; *ν*_2_—quantity of AB-17-8, mol; *n*_1_, *n*_2_—molar quantity of the cation exchanger and anion exchanger at certain molar ratio in the interpolymer pair.
$$Q=\frac{m_{sorbed}}{m_{sorbent}}$$
where *m_sorbed_*—mass of sorbed Nd^3+^ or Sc^3+^ ions, mg; *m*—ion-exchanger weighed portion (if there are two ion-exchange resins in the corresponding salt solution; this value is determined as the sum of total weight of each ion-exchanger), g.

### 2.2. Methods

#### 2.2.1. Determination of Electrochemical Properties

For the determination of the electric conductivity, the Expert 002 conductometer (Econics-expert, Moscow, Russian Federation) was used. Measurement of pH values was performed on pH-meter 827 Metrohm (Metrohm, Herizau, Switzerland). A study of these electrochemical properties of aqueous medium and neodymium and scandium sulfates solutions provides possibility for prediction of appearance of areas of high ionization of polymer structures during a remote interaction.

#### 2.2.2. Neodymium and Scandium Ions Concentration Determination

The KFK-3KM (Unico Sys, Saint-Petersburg, Russian Federation) Photocolorimeter was used to determine the optical density (for further calculation of Nd^3+^ and Sc^3^ ions concentration). In addition, the concentration was determined on an ICP-OES spectrometer 8300 ICP-OES (Perkin Elmer, Waltham, MA, USA). The importance of the neodymium and scandium concentration is the fact that the concentration is the basis for the calculation of the mentioned above sorption properties.

#### 2.2.3. Fourier-Transform Infrared Spectra of the Sorbents 

FTIR-spectra of the initial ion-exchange structures and the interpolymer system was obtained with using a NICOLET 5700 spectrophotometer (Thermo Fischer scientific, Waltham, MA, USA).

#### 2.2.4. Thermogravimetric Curves of the Sorbents

The LabSYS evo TG/DTA 1600 (Setaram, Caluire, France) thermoanalyzer was used for the determination of the thermogravimetric properties of the initial ion-exchange structures and the interpolymer system. TGA/DTA conditions: dynamic mode; interval of temperature—20–630 °C heating rate 10 °C/min; nitrogen atmosphere.

## 3. Results and Discussion

For a clear study of the influence of the certain rare-earth metal ionic radius (neodymium or scandium in case of the study) on the phenomenon of the remote interaction and, as a consequence, sorption process of ions of these metals it is necessary to study behavior of individual ion-exchangers and interpolymer systems on their basis in an aqueous medium for obtaining a clear understanding of ionization process during mutual activation of the polymer structures.

### 3.1. Electrochemical Behavior of Individual Ion-Exchange Resins and Interpolymer System in an Aqueous Medium

Specific electric conductivity and pH dependencies of an aqueous medium is dependent on Amberlite IR-120 molar amount and from time are presented in Figure 1: (a) and (b), respectively. The dissociation process of functional groups of the mentioned ion-exchanger leads to an increase of electric conductivity with time, where the direct result of the dissociation process is the formation of charged functional groups and protons. As seen from Figure 1, an increase of the molar amount of the ion-exchanger provides increase of specific electric conductivity, wherein a significant increase (over 70%) of the electrochemical parameter is observed with increase of molar amount of Amberlite IR120 from 5 to 6 mol. Such a strong increase is a consequence of dissociation of excess functional groups in the presence of 6 mol Amberlite IR120 in an aqueous medium. A significant part of the functional groups of the ion-exchange resin undergoes dissociation, leading to a strong increase in electric conductivity. A change in the concentration of H^+^ ions in the presence of the Amberlite IR120 ion-exchanger points to release of protons with time due to dissociation of functional groups of the cation exchanger, which, in turn, is evidenced by the growth in the amount of hydrogen ions in the aqueous medium. 

In accordance with data on a specific electric conductivity (the parameter increases with time), the peaks indicating an increase in conductivity are observed at 4 and 6 moles of the Amberlite IR120, wherein the strongest release of protons occurs in the presence of 6 moles of the ion-exchanger. Electric conductivity and pH dependencies of an aqueous medium is dependent on molar amount of AB-17-8 and time are presented in Figure 1: (c) and (d), respectively. Increase of electric conductivity in this case (presence of AB-17-8 ion-exchange resin) occurs due to dissociation of water molecules to H^+^ ions and OH^−^ groups. An anion exchanger binds protons from the water medium, which, in turn, provides additional dissociation (in accordance with Le-Chatelier’s principle) of water molecules (due to a shift in electrochemical equilibrium to the right). Each increase of the amount of the ion exchanger provides an increase in electric conductivity over 20%. From the results, the character of the hydrogen ions concentration change in the presence of the AB-17-8 ion-exchanger in an aqueous medium showed that in the beginning of the experiment (after 5 min after start of interaction), there was a strong increase in pH caused by the association of protons present in the aqueous medium. The subsequent pH decrease phenomenon points to a higher concentration of hydrogen ions in the water medium with return to initial pH values (equilibrium). Nevertheless, the pH remains increased at an amount of 6 moles of AB-17-8, which may indicate the ionized state of the anion exchanger due to an excess of ionized heteroatoms and released protons.

The electrochemical behavior of the individual cation exchange resin and anion exchange resin in an aqueous medium differ from each other (Figure 1). An increase in the share of the cation exchanger Amberlite IR120 in aqueous medium leads to increase of specific electric conductivity with time, with similar decrease of pH indicating to dissociation of functional groups (strong dissociation occurs in the presence of six moles of the cation exchanger). In the presence of the anion exchange structure, the specific electric conductivity decreases with time, which indicates the binding of free protons formed due to dissociation of water molecules by the heteroatom of AB-17-8. This, in turn, causes additional dissociation of free water molecules. An increase in the molar amount in the anion exchanger leads to an increase in the values of the parameter. A decrease in the pH values at molar amount of AB-17-8 from 1 to 4 mol indicates that dissociation prevails over the association process; further increase of molar amount to 5 and 6 mol leads to decrease of H^+^ concentration in water medium, which can evidence an increase of association of protons.

A specificity of the ionization process in the interpolymer system of the Amberlite IR120-AB-17-8 is the absence of a counterion in ionized groups as a result of interpolymer interactions with further mutual activation of polymers, and the formation of uncompensated charges along the polymer chains on the initial polymers. Uncompensated charges are formed as a result of the protons’ release during the dissociation process of the functional groups of the cation exchange resin of Amberlite IR120 and association of these mobile ions by the heteroatoms of the anion exchange resin AB-17-8 in an aqueous medium, wherein the charge density of AB-17-8 is limited by the degree of dissociation of the Amberlite IR120. The consequence of these interactions is the ionization of both polymer structures with the formation of similarly charged groups on the links of internode chains without counterions.

Ionization of the ion-exchangers occurs in two stages. First stage—hydration of polymer links; second stage—unfolding of one polymer structure under the action of the second polymer molecule, giving the total solution either H^+^ or OH^−^. Unfolding provides opening the links that form intermolecular bonds. As a result of the remote interaction, functional groups are formed without counterions, which is facilitated by the destruction of intra-salt bonds stabilized by hydrophobic interactions, since divinylbenzene in the composition of the anion exchanger is a hydrophobic fragment. High ionization state of each macromolecule is due to the increase of ionization areas and relaxation areas during mutual activation. Mutual activation is a phenomenon that provides is transition of the cation and the anion exchangers in the interpolymer system from stationary state into a more reactive state.

The specific electric conductivity of water medium is significantly influenced by presence of the studied interpolymer system (Amberlite IR120-AB-17-8) (Figure 2a). Obtained data points to the influence of remote interaction phenomenon on conformational changes of the initial ion-exchange resins, such changes in structure of the macromolecules in interpolymer pairs lead to significant changes of their electrochemical properties. Increase of the electrochemical parameter in the presence of the individual ion-exchangers (molar ratios 6:0 and 0:6) can be explained as follows: amount of protons is released in water medium due to dissociation of functional groups of the cation exchange resin; additional dissociation of water molecules with the release of hydrogen ions in the presence of AB-17-8 is caused by decrease of protons amount during ionization of anion exchanger. Low values of electric conductivity for all times of interaction are observed at ratio Amberlite IR120:AB-17-8 = 3:3. High values of electric conductivity are observed at ratios of Amberlite IR120:AB-17-8 = 4:2 and 2:4. Character of change concentration of hydrogen ions in the presence of the interpolymer system Amberlite IR120-AB-17-8 is dependent on molar ratios and time is presented in Figure 2b. In the presence of the individual structure, the Amberlite IR120’s decrease in pH indicates a dissociation of the functional groups, while increase of pH in the presence of individual AB-17-8 points to a proton binding by a heteroatom. High values of hydrogen ions concentration in the interpolymer pair Amberlite IR120:AB-17-8 = 4:2 may indicate that there is a predomination of dissociation over association. Low concentration of protons is observed at ratio 5:1, compared to electric conductivity data this ratio can be named as the high ionization area of the initial macromolecules.

Remote interaction phenomenon significantly affects the electrochemical properties of an aqueous medium. The presence of the interpolymer pairs provides significant changes of the initial electrochemical parameters, particularly the values of specific electric conductivity in an interpolymer system (molar ratios of Amberlite IR120:AB-17-8 from 5:1 to 1:5) are comparatively lower with individual ion-exchangers (molar ratios 6:0 and 0:6). This data points to the existence of intermolecular changes in the structure of the ion-exchangers, wherein the occurrence of ionization of each macromolecular structure leads to the breaking of intermolecular bonds. The not particularly high values of proton concentration in the integral pairs of Amberlite IR120:AB-17-8 = 5:1 points to prevalence of the proton association process over the functional groups’ dissociation process; as a consequence, additional groups undergo dissociation and, at final ionization, the degree of both ion exchangers is at a maximum.

### 3.2. Sorption of Neodymium and Scandium Ions by the Interpolymer System Amberlite IR120-AB-17-8

#### 3.2.1. Electrochemical Behavior of the Interpolymer System Amberlite IR120-AB-17-8 in Neodymium and Scandium Sulphates Solutions

Specific electric conductivity and pH of neodymium sulfate solution’s independence from the ion-exchange resins’ molar ratios and interaction times are presented in Figure 3a,b, respectively. The “long-range effect” provides significant changes in electric conductivity of the neodymium sulfate solution in the presence of the Amberlite IR120-AB-17-8 interpolymer system, particularly the electrochemical parameter increases with time for all molar ratios, except 0:6. In this case, some part of the present amount of protons undergoes binding by heteroatom of the individual anion exchanger AB-17-8. Individual cation exchange resins of the Amberlite IR120 during ionization in the salt solution undergoes a dissociation of the functional groups with a subsequent release of protons in the reactive medium—this is the main reason for the increase in electric conductivity. 

The macromolecular polymer structures in the interpolymer pairs undergo significant conformational changes due to mutual activation, and, as a result of this phenomenon, are transferred into a highly ionized state. Relatively high values of conductivity can be observed at ratio 5:1 during remote interaction. A linear, slight decrease of pH values of the rare-earth metal salt in the presence of individual Amberlite IR120 macromolecules is a direct result of dissociation of functional groups to H^+^ and OH^−^ groups. The opposite situation is observed in the presence of the individual anion-exchanger in the AB-17-8–the decrease in pH evidences the ionization of the ion-exchange resin by a portion of the protons along with H_2_O molecule formation. 

It can be said that the interpolymer system’s components undergo ionization during remote interaction. Obtained data points to decreases in pH values with time for all molar ratios in the interpolymer system of the Amberlite IR-120-AB-17-8. A high concentration of hydrogen ions can be observed at ratio 5:1, it can be concluded that protons concentration growth occurs due to additional dissociation of functional groups of cation exchanger during remote interaction. Specific electric conductivity and pH of scandium sulfate solution is dependent on the ion-exchange resins’ molar ratios and interaction time are presented in Figure 3a,b. An increase of conductivity values with time occurs due to proton release. The mutual activation of the initial ion-exchange resins in the interpolymer system lead to an ionization increase with a further transition of the macromolecular structures from a stationary state to a more reactive state. High conductivity values are observed at ratios 5:1 and 2:4 from 4.5 h up until 24 h of interaction. The lowest values of the electrochemical parameter are observed at ratio 3:3 for all time of interaction. Protons concentration increases with time in the presence of the individual cation exchanger Amberlite IR120 due to the dissociation of functional groups. A decrease in the concentration of hydrogen ions in the presence of individual anion-exchanger points to the binding of protons formed during H_2_O molecules dissociation, by heteroatom of the anion exchange structure AB-17-8. Decrease of pH in the interpolymer system is observed at ratios of Amberlite IR-120:AB-17-8 from 6:0 to 5:1 and from 3:3 to 0:6. Low values of concentration of hydrogen ions are the consequence of ionization of polybias with further release of OH^−^ groups, which bind with H^+^ ions resulting in the formation of water molecules. Remote interaction leads to additional dissociation of functional groups of the cation exchanger due to binding of the released protons.

The impact of neodymium and scandium ions’ ionic radius on sorption by the interpolymer system of the Amberlite IR120-AB-17-8 can be observed on the above-mentioned electrochemical curves (Figure 3). The sorption of neodymium ions is accompanied by increase of specific electric conductivity for all molar ratios (except 0:6—in the presence of individual AB-17-8 electric conductivity decreases with time) until 40 h, after that time there is a slight decrease of the parameter. There are significant changes in structure of the ion-exchange resins at the scandium ions’ soprtion in integral pairs. Electric conductivity increases with time. 

However, the lowest values of the parameter are observed at ratio 3:3 for all time of interaction, while protons concentration increases with time (for all ratios except 0:6). Decrease of pH values in both cases (sorption of neodymium and scandium ions) points to functional groups additional dissociation wherein there is a prevalation of dissociation over association process. There is a difference in behavior of AB-17-8 in the corresponding salt solutions: in neodymium sulfate solution, the pH increases; in scandium sulfate solution, the pH decreases with time. Such difference is a direct result of formation of coordination bonds–sorption of neodymium ions is complicated due to big radius of the ion owing to this fact competing reaction of protonization of heteroatoms also takes place.

#### 3.2.2. Sorption Properties of the Interpolymer System Amberlite IR120-AB-17-8 in Relation to Nd^3+^ and Sc^3+^ Ions

Change of concentration of Nd^3+^ (a) and Sc^3+^ (b) ions during sorption by the interpolymer system is presented on Figure 4 is dependent on ion-exchange resins’ molar ratios and from interaction time. A decrease in the concentration of the REM with time indicates the occurrence of the soprtion of neodymium and scandium ions by the interpolymer system. Strong sorption of neodymium occurs at 0.1 h, therein the concentration of the metal significantly decreases (from 100 mg/L to 90.31 mg/L) at a ratio of the Amberlite IR120:AB-17-8 = 4:2, when the decrease of the rare-earth element concentration due to sorption for individual Amberlite IR120 is from 100 mg/L to 98.90 mg/L and AB-17-8 is from 100 mg/L to 99.20 mg/L. Further sorption is accompanied by significant changes in structure of the ion-exchangers at ratio Amberlite IR120:AB-17-8 = 4:2 resulting to release of some part of previously sorbed neodymium ions back into sulfate solution due to relaxation effect. Beginning from 4.5 h, the exact maximum of sorption is observed at ratio 5:1. As it was mentioned earlier, this ratio is an area of maximum ionization of the cation and anion exchangers. The most significant increase of sorption is observed at this ratio at 48 h of interaction, initial concentration of neodymium is decreased from 100 mg/L to 57.68 mg/L, while the decrease for Amberlite IR120 is from 100 mg/L to 61.59 mg/L and for AB-17-8 is from 100 mg/L to 77.98 mg/L. The sorption of scandium ions is accompanied by significant decrease of the metal’s concentration at ratio 5:1 from the starting moment of the sorption (0.1 h) comparatively with individual ion-exchange structures Amberlite IR120 and AB-17-8. At this time concentration of the metal decreases from 100 mg/L to 96.64 mg/L in the presence of Amberlite IR120 and from 100 mg/L to 97.30 mg/L in the presence of AB-17-8, it should be noted that the most intense sorption occurs at ratio 5:1 (concentration decreases from 100 mg/L to 90.90 mg/L). Beginning from 4.5 h, the exact peaks of low concentration appear at ratios 5:1 and 6:0. The intense sorption of the rare-earth metal occurs during 48 h, wherein maximum sorption peaks of scandium appear at 48 h at ratio 1:5; the concentration of Sc^3+^ ions decreases from 100 to 61.94 mg/L while the decrease for Amberlite IR120 is from 100 mg/L to 65.03 mg/L, and for the AB-17-8 is from 100 mg/L to 76.21 mg/L.

The extraction degree of Nd^3+^ (a) and Sc^3+^ (b) ions of the interpolymer system Amberlite IR120-AB-17-8 is presented in Figure 5 with a dependence on the ion-exchangers molar ratios in terms of time. The extraction degree of Nd^3+^ ions increases with time, wherein it is observed that ratio where the parameter has maximum values is Amberlite IR120:AB-17-8 = 5:1. A significant increase of extraction degree occurs at the time interval from 24 h to 48 h; the increase of sorption is from 27.43% to 42.32%. For individual Amberlite IR120, the increase is from 25.48% to 38.41%, for AB-17-8—from 12.00% to 22.02%. The sorption parameter also increases with time in the case of Sc^3+^ ions sorption; the maximum values of extraction degree can be observed at 48 h of interaction. A strong increase of the sorption parameter is observed at ratio 5:1, wherein sorption degree is 38.06%, while extraction degree is 34.97% (Amberlite IR120) and 23.79% (AB-17-8). An analysis of scandium sorption time of provides conclusion that maximum sorption area of Sc^3+^ ions is ratio 5:1.

Polymer chain binding degree (relatively to Nd^3+^ (a) and Sc^3+^ (b)) of the interpolymer system Amberlite IR120-AB-17-8 is presented in Figure 6 is dependent on the ion-exchange resins’ molar ratios of and interaction time. Significant increase of binding degree (in relation to neodymium ions) in the interpolymer system is observed from the moment of sorption beginning (0.1 h), binding degree at ratio Amberlite IR120:AB-17-8 = 5:1 is 0.39%, while it is 0.08% for Amberlite IR120 and 0.07% for AB-17-8. Subsequent strong increase comparatively with individual ion-exchangers is observed at this ratio at 0.5 h of interaction, the parameter is 0.72% for Amberlite IR120:AB-17-8 = 5:1, 0.27% for Amberlite IR120 and 0.52% for AB-17-8. The emote interaction in this interpolymer pair leads to areas of high binding of neodymium ions at 15. The binding degree at 15 h is 1.58%, while it is 0.92% for Amberlite IR120 and 0.82% for AB-17-8. The end of sorption (40 h) is accompanied with area of strong difference in values of polymer chain binding degree, for Amberlite IR120:AB-17-8 = 5:1 binding degree is 3.06%, while it is 2.10% for Amberlite IR120 and 1.06% for AB-17-8. 

The maximum values of the parameter are reached at 48 h; it is 3.23% for the interpolymer pair; 2.89% for Amberlite IR120; 1.80% for AB-17-8. Polymer chain binding degree (relatively to scandium ions) of the interpolymer system Amberlite IR120-AB-17-8 increases with time, the most significant increase of the parameter is observed at Amberlite IR120-AB-17-8 molar ratio 5:1 for all time of interaction. At the initial moment of sorption (0.1 h) the difference in values of binding degree of the interpolymer ratio and individual ion-exchangers is more than 2 times—1.06% for Amberlite IR120:AB-17-8 = 5:1, 0.39% for Amberlite IR120 and 0.34% for AB-17-8. Further increase of the sorption parameter occurs slightly up to 15 h, at this time binding degree is 2.01% for Amberlite IR120:AB-17-8 = 5:1, 1.19% for Amberlite IR120 and 1.44% for AB-17-8. Binding degree is 4.16% for this ratio at 40 h of remote interaction, absence of remote interaction phenomenon in case of presence of individual Amberlite IR120 and AB-17-8 leads to the fact that binding degree is relatively low in this case: 2.97% for Amberlite IR120 for and 1.60% for AB-17-8. Further increase of binding degree (up to 48 h) is not so strong in the interpolymer pair Amberlite IR120:AB-17-8 = 5:1, it is 4.43%, while the parameter is 4.01% for Amberlite IR120 and 2.96% for AB-17-8.

Effective dynamic exchange capacity (relatively to neodymium (a) and scandium (b) ions) of the interpolymer system Amberlite IR120-AB-17-8 is presented on Figure 7 is dependent on the ion-exchange resins’ molar ratios and the interaction time. The sorption of Nd^3+^ ions is accompanied with strong increase of exchange capacity in the interpolymer pair Amberlite IR120:AB-17-8 = 5:1 from the beginning of sorption (0.1 h)—it is 215.83 mg/g; while the capacity is 45.83 mg/g for Amberlite IR120 and 33.33 mg/g for AB-17-8. 

Another increase in the soprtion parameter is observed at 0.5 h; for the interpolymer pair, it is 391.25 mg/g; for Amberlite IR120—149.17 mg/g; for AB-17-8—265.83 mg/g. Further significant increase of exchange capacity can be observed at 17 h–it is 862.50 mg/g for Amberlite IR120:AB-17-8 = 5:1; 579.17 mg/g for Amberlite IR120; 284.58 mg/g for AB-17-8. The subsequent sorption of neodymium ions leads to area of strong increase of capacity at 40 h of interaction: exchange capacity is 1670.83 mg/g for Amberlite IR120:AB-17-8 = 5:1; 1161.67 mg/g for Amberlite IR120; 541.67 mg/g for AB-17-8. After that time, the growth of the sorption parameter in the interpolymer pair is not very intense; it is 1763.33 mg/g for Amberlite IR120:AB-17-8 = 5:1; 1600.42 mg/g for Amberlite IR120; 917.50 mg/g. Exchange capacity (in relation to scandium ions) of the interpolymer system Amberlite IR120-AB-17-8 increases with time. From obtained data, it was seen that The maximum values of exchange capacity are reached at ratio Amberlite IR120:AB-17-8 = 5:1. Strong increase of capacity as a result of high ionization is observed at 0.1 h of sorption; capacity is 379.17 mg/g for this ratio; 140.00 mg/g for Amberlite IR120 and 112.50 mg/g for AB-17-8. Up to 17 h of interaction, the parameter increases; at this time significant increase is observed—it is 815.83 mg/g for the interpolymer pair, while it is 445.83 mg/g for Amberlite IR120 and 500.00 mg/g for AB-17-8. High ionization at 24 h of the remote interaction provides strong difference in values of capacity–1320.83 mg/g for Amberlite IR120:AB-17-8 = 5:1; 891.25 mg/g for Amberlite IR120; 505.42 mg/g for AB-17-8. At 48 h (end of sorption process), the effective dynamic exchange capacity is 1585.83 mg/g for the interpolymer pair; 1457.08 mg/g for Amberlite IR120 and 991.25 mg/g for AB-17-8.

Such a strong increase of the studied sorption parameters is the direct result of formation of optimal conformation for soprtion of Nd^3+^ and Sc^3+^ ions during mutual activation of the initial ion-exchangers for sorption of neodymium and scandium ions.

### 3.3. Initial Sorption Properties Growth Due to Remote Interaction 

The obtained data on sorption of neodymium and scandium ions (Figure 4, Figure 5, Figure 6 and Figure 7) showed that maximum sorption of the REM observed at ratio Amberlite IR120:AB-17-8 = 5:1. Due to this fact, this ratio was selected for comparison with individual ion-exchangers.

Table 1 represents the growth of extraction degree of neodymium ions of the interpolymer pair comparatively with individual polymer structures of Amberlite IR120 and AB-17-8. The strongest growth is observed at 0.1 h of interaction; it is 370.91% comparatively with the cation exchange resin (Amberlite IR120) and 547.50% comparatively with the anion exchange resin (AB-17-8). At 0.5 h of interaction, the growth of sorption degree is 162.29% for Amberlite IR120 and 47.18% for AB-17-8. The growth of the sorption parameter is 73.62% for the cation exchanger and 65.53% for the anion exchanger at 1.5 h. The time of 6.5 h is accompanied with growth of the parameter 70.83% for the cation exchanger and 75.61% for the anion exchanger. Times of 15, 17 and 20 h provide significant growth of the parameter for AB-17-8; the growth is 106.70%; 203.07% and 290.62%, respectively. For the cation exchange, the resin from these time intervals’ high growth of extraction degree is observed only at 15 h; it is 68.73%. The intense growth for both ion-exchangers occurs at 40 h of interaction, the extraction degree increases up to 43.83% comparatively with the cation exchanger and up to 208.46% comparatively with the anion exchanger. At the end of sorption (48 h), growth is insignificant relative to Amberlite IR120 (10.18%) and strong comparatively with AB-17-8 (92.19%). In addition, the growth of extraction degree of scandium ions of ratio 5:1 comparatively with individual ion-exchange structures is presented in Table 1. The high ionization of both macromolecular structures in the interpolymer pair in initial moment of interaction provides significant increase of sorption degree (the parameter increases up to 170.83% in relation to Amberlite IR120 and 237.04% in relation to AB-17-8). The average increase of the extraction degree at the time interval from 0.5 h to 15 h is over 47% relatively to the cation exchanger and almost 40% in relation to the anion exchanger. Strong growth is observed at 17 h: 82.99% relatively to Amberlite IR120 and 63.17% relatively to AB-17-8. Further interaction (20–48 h) with scandium sulfate leads to decrease of sorption degree growth relatively to the cation exchange resin (66.87%–48.20%–38.22%–8.84%) and to increase of the sorption parameter in relation to the anion exchange resin (108.31%–161.34%–178.60%–59.98%).

The values of growth of polymer chain binding degree (in relation to neodymium ions) of the interpolymer system Amberlite IR120:AB-17-8 = 5:1 comparatively with individual ion-exchange resins presented in Table 2. Significant growth of polymer chain binding degree (in relation to neodymium ions) at molar ratio Amberlite IR120:AB-17-8 = 5:1 is observed at 0.1 h of interaction, the parameter increases up to 377.01% in comparison with Amberlite IR120 and increases up to 505.52% in comparison with AB-17-8 at this time. A strong increase for both ion-exchangers is observed at 0.5 (165.69% and 37.64% respectively) and 1.5 h (75.87% and 54.80%). An increase of 42.42% and 98.05% is observed at 4.5 h, 73.05% and 64.22% at 6.5 h. Conformational changes in the structure of initial ion-exchangers at time interval 15-20 h leads to decrease of growth of sorption degree in comparison with Amberlite IR120 (70.92%–50.85%–17.17%) and to increase of growth of sorption degree in comparison with AB-17-8 (93.30%–183.43%–265.30%). In addition, the strong growth of the parameter is observed at 24 h of interaction compared to the anion exchanger (113.76%), while it was only 9.05% comparative to the cation exchanger. The final significant increase occurs at 40 h of remote interaction—it is 45.70% in comparison with Amberlite IR120 and 188.46% in comparison with AB-17-8. Final point of sorption—48 h of interaction is accompanied with growth of 11.61% and 79.73%, respectively. Growth of polymer chain binding degree (in relation to scandium ions) of the interpolymer pair comparatively with individual ion-exchange resins is presented in Table 2. The most significant growth of binding degree is observed at 0.1 h of remote interaction, it is 174.34% in relation to Amberlite IR120 and 215.19% in relation AB-17-8. A further increase (up to 15 h) is not intense—about 40% in relation to both macromolecules. The time area from 17 h to 48 h is accompanied with a slight decrease of binding degree growth from 85.36% to 10.25% relative to Amberlite IR120 and an increase of the parameter from 52.59% to 160.54% (with a decrease of growth at 48 h to 49.61%) relative to AB-17-8.

Table 3 presents values of growth of the effective dynamic exchange capacity of the interpolymer system Amberlite IR120:AB-17-8 = 5:1 comparatively with individual ion-exchangers Amberlite IR120 and AB-17-8. The high ionization in the initial moment of time (0.1 h) provides a strong increase of exchange capacity; the growth is 408.00% comparatively with Amberlite IR120 and 602.25% with AB-17-8. From 0.5 to 2.5 h the decrease of growth, comparatively with Amberlite IR120, it is more than 4 times (178.52%–80.98%–40.89%), while in comparison with AB-17-8, it undergoes increase with subsequent decrease more than 2 times (51.90%–72.08%–21.06%). 

A strong growth of the parameter comparatively with AB-17-8 occurs at time interval 15–20 h (117.37%–223.38%–319.68%), while in comparison with Amberlite IR120 the growth decreases (75.61%–53.81%–17.23%). Significant growth is observed at 17 and 40 h of remote interaction. At 17 h, the exchange capacity in the interpolymer pair increases up to 53.81% comparatively with Amberlite IR120 and up to 223.38% comparatively with AB-17-8. At 40 h the parameter increases up to 48.21% and 229.31% respectively. At the end of sorption experiment (48 h), growth is significant in comparison with AB-17-8; it is 101.41% and insignificant in comparison with Amberilte IR120; it is 11.20%. Values of growth of the exchange capacity (relative to Sc^3+^ ions) of ratio 5:1 in comparison with the individual ion-exchange resins are also presented in Table 3. The strongest growth of the parameter was seen at 0.1 h of interaction—the growth relatively to Amberlite IR120 is 187.92%; relatively to AB-17-8—260.74%. High growth values of binding degree are observed at 17 h: growth is 91.29% relative to the cation exchanger and 69.48% relative to the anion exchanger. Further interaction (up to 40 h) is accompanied with a growth decrease (from 73.56% to 42.05%) in relation to Amberlite IR120, while there is an increase of parameter growth in relation to AB-17-8 (from 119.14% to 196.46%). Growth of binding degree at 48 h of remote interaction is insignificant—9.72% relative to Amberlite IR120 and 65.98% relative to AB-17-8.

The average growth of the studied sorption properties (mean value of the total sum of ω(η), ω(θ), ω(Q) from sorption of Nd and Sc ions) of Amberlite IR120 (a) and AB-17-8 (b) from ratio 5:1 comparatively with initial cation-exchange resin Amberlite IR120 and anion exchange resin AB-17-8 from sorption of the REM ions is presented in Figure 8. Significant growth of the sorption properties of Amberlite IR120 from ratio 5:1 comparatively with individual Amberlite IR120 for both REM is observed at 0.1 h; further interaction leads to decrease of the growth up to 2.5 h. After that, a remote interaction provides a slight increase of the growth beginning from 4.5 h up to 15 h for both metals. After 15 h of interaction, the character of growth change is straightforward for Nd and Sc ions. In the case of Nd ions, the sorption growth decreases up to 24 h with further increase up to 40 h, while sorption of Sc ions is accompanied with sharp growth increase up to 17 h, with further growth decrease up to 40 h. The sorption of the both REM ions occurs simultaneously in the time interval from 40 to 48 h; this is characterized by a decrease of sorption property growth for both cases. Growth of AB-17-8 from ratio 5:1 sorption properties from sorption of Nd and Sc ions is accompanied with strong increase at 0.1 h with further decrease up to 2.5 h. Subsequent interactions (up to 20 h) of the ion exchanger with the salt solutions is accompanied with strong growth of the sorption properties, where growth increase for Nd ions sorption is higher in comparison with Sc ions sorption. The character of further growth change is different for these cases: growth decreases for Nd ions sorption up to 24 h after that it slight increases up to 40 h, while sorption of Sc ions is accompanied with slit increase of growth up to 40 h. An almost-similar situation was observed at an interval 40 to 48 h—growth decreased for both cases.

### 3.4. FTIR and TGA Characterisation of the Cation Exchange Resin and Anion Exchange Resin before and after Neodymium Ans Scandium Sorption

Obtained data (Figure 4, Figure 5, Figure 6 and Figure 7) showed that the maximum sorption of Nd^3+^ and Sc^3+^ occurs at ratio of Amberlite IR120:AB-17-8 = 5:1 for both studied REM. Owing to this fact, this ratio of the cation and anion exchangers was taken for further FTIR and TGA study. The macromolecular structures are taken from the time of interaction 0.5 h (sorption of Nd^3+^ ions) and from the time of interaction 15 h (sorption of Sc^3+^ ions) due to the fact that difference of the sorption parameters is at a maximum at these times.

#### 3.4.1. FTIR Spectra of the Initial Ion-Exchangers and the Interpolymer Pair Amberlite IR120:AB-17-8 = 5:1

FTIR spectra of initial Amberlite IR120 (without sorbed Nd^3+^ ions), individual Amberlite IR120 (with sorbed Nd^3+^ ions) and Amberlite IR120 from the interpolymer pair Amberlite IR120:AB-17-8 = 5:1 (with sorbed Nd^3+^ ions) are presented on Figure 9a. FTIR spectra initial Amberlite IR120 (without sorbed Sc^3+^ ions), individual Amberlite IR120 (with sorbed Sc^3+^ ions) and Amberlite IR120 from the interpolymer pair Amberlite IR120:AB-17-8 = 5:1 (with sorbed Sc^3+^ ions) are presented on Figure 9b. Changes in absorbance after sorption of the REM (wavenumbers interval 1330–1375 cm^−1^, which correspond to S = O stretching in sulphonic acid) are presented in the figure, wherein the values of absorbance in the case of the interpolymer pair are higher. This, in turn, points to a more intense sorption of Nd^3+^ and Sc^3+^ ions.

Figure 10a presents FTIR spectra of initial AB-17-8 (without sorbed Nd^3+^ ions), individual ion-exchanger AB-17-8 (with sorbed Nd^3+^ ions) and AB-17-8 from ratio 5:1 (with sorbed Nd^3+^ ions). Figure 10b presents comparative analysis of FTIR spectra of initial AB-17-8 (without sorbed Sc^3+^ ions), individual ion-exchanger AB-17-8 (with sorbed Sc^3+^ ions) and AB-17-8 from ratio 5:1 (with sorbed Sc^3+^ ions). Ionizations of the heteroatom of the cation-exchanger and anion exchanger is the reason for the absorbance changes due to the sorption of Nd^3+^ and Sc^3+^ ions. Sorption of both REM leads to changes in the structure of the anion exchange resin AB-17-8, which is observed in the FTIR spectra. Absorbance increases in wavenumber interval 3000–2800 cm^−1^ due to N-bond stretching due to the sorption of neodymium and scandium ions by the macromolecular structures.

#### 3.4.2. TGA Analysis of the Initial Ion-Exchangers and the Interpolymer Pair Amberlite IR120:AB-17-8 = 5:1

TGA curves of neodymium sulfate and scandium sulfate are presented in Figure 11 a,b. Thermal destruction of neodymium sulfate takes place from 140 °C to 200 °C with further decomposition up to 310 °C. It is indicated by a strong decrease of mass loss (the parameter decreases from 100% to 86%). It should also be noted that the rate of mass loss decreases (from 140 °C to 190 °C) with a further increase (from 190 °C to 310 °C) at this temperature area. Thermo destruction of the scandium salt occurs from 90 °C to 190 °C with further decomposition up to 320 °C. It is evidenced by mass loss (the parameter decreases from 100% to 94.5%). At the same time (process of thermal decomposition of scandium sulfate) rate of mass loss initially decreases (from 80 °C to 170 °C) with a subsequent increase (from 170 °C to 280 °C).

A comparison of thermal destruction curves (mass loss (a) and rate of mass loss (c)) of individual Amberlite IR120 (without sorbed Nd^3+^ ions), Amberlite IR120 from ratio 6:0 (with Nd^3+^ ions) and Amberlite IR120 (with Nd^3+^ ions) from ratio 5:1 is presented in Figure 12. From the obtained data, it was seen that the mass loss of the three samples of Amberlite IR120 occurs almost simultaneously up to 150 °C. Further mass loss occurs faster for Amberlite IR120 6:0 and Amberlite IR120 1:5 due to the fact of presence of neodymium ions in the macromolecular matrix, wherein the rate of mass loss for individual ion-exchanger is lower in comparison with Amberlite IR120 from the interpolymer pair due to a lesser amount of sorbed metal in the ion-exchange structure. Thermal decomposition of Amberlite IR120 occurs up to 400 °C. The temperature area from 400 °C to 625 °C can be named as stabilization area with the indication of the process ending. The figure also presents a comparison of TGA curves (mass loss (b) and rate of mass loss (d)) of the following macromolecular structures: individual Amberlite IR120 (without sorbed Sc^3+^ ions), Amberlite IR120 from ratio 6:0 (with Sc^3+^ ions) and Amberlite IR120 (with Sc^3+^ ions) from molar ratio 5:1. The obtained results show that the process of thermal destruction (particularly mass loss) occurs almost simultaneously until 200 °C. The temperature area from 300 °C to 450 °C is the thermal destruction area for all of the samples. Further heating of these samples leads to a stabilization area, which indicates the end of the process.

Figure 13 represents a comparison of the thermal destruction curves (mass loss (a) and rate of mass loss (c)) of individual AB-17-8, AB-17-8 from ratio 6:0 (with Nd^3+^ ions) and AB-17-8 from ratio 5:1 (with Nd^3+^ ions) from time of interaction 0.5 h. A difference in mass loss of the three chosen samples of AB-17-8 is observed from 210 °C; before this, temperature it occurs almost simultaneously. Further thermal decomposition is accompanied with strong mass loss from 380 °C to 450 °C, where the appearance of differences in the values of the mass loss is due to neodymium ions sorption. Further decomposition (in temperature interval from 450 °C to 625 °C) is accompanied by slight decrease of the mass loss due to stabilization of thermal destruction process. The figure represents comparison of thermal destruction curves (mass loss (b) and rate of mass loss (d)) of individual AB-17-8, AB-17-8 from ratio 6:0 (with Sc^3+^ ions) and AB-17-8 from ratio 5:1 (with Sc^3+^ ions) from time of interaction 15 h. Thermal destruction process occurs almost the same for the three chosen samples of AB-17-8 up to 225 °C. Further destruction (from 225 °C to 450 °C) is accompanied with differences in the values of the mass loss is due to scandium ions sorption. After 450 °C, a slight decrease of mass indicates a stabilization of the process of thermal decomposition.

## 4. Conclusions

Based on the obtained results of the study, it is possible to predict the highly ionized state of the initial cation and anion exchangers during their remote interaction in interpolymer systems. Sorption methods applied in hydrometallurgy can be improved by the application of the “long-range effect” on industrial ion-exchangers. Based on this, new sorption technologies and methods can be developed. Highly ionized ion-exchange resins can be successfully used as independent sorbents for the selective sorption of REM ions. The conducted studies cover the sorption of neodymium and scandium, but it is known that all REM are almost similar in chemical properties and it is possible to predict that remote interaction phenomenon can lead to a significant increase in the initial sorption properties of the macromolecules for the selective sorption of the studied rare-earth metals. The remote interaction effect can be successfully used for the effective modification of industrial ion-exchangers for the effective sorption of rare-earth metals.

## Data Availability

The raw/processed data required to reproduce these findings cannot be shared at this time due to technical or time limitations.

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
