# Peer review of "Impact of Neodymium and Scandium Ionic Radii on Sorption Dynamics of Amberlite IR120 and AB-17-8 Remote Interaction"

_materials, 2021, doi:10.3390/ma14185402_

Round 1

Reviewer 1 Report

The presented manuscript covers the effects of a co-polymer ion exchange system (based on Amberlite resins) and their effects on trivalent Sc and Nd sorption.

A few notes from the text that I'd like to see addressed by the authors:

  • Line 38 - Scandium is a very rare element and has very few niche uses, outside of alloys in limited applications, given worldwide produciton is < 20 tons/year. Nd is far more useful and processed on a far larger scale. Please correct this and in lines 47-64. This paragraph can likely be condensed and merged with the one discussing Nd applicaitons. Statement of total worldwide production (using latest values of mass, and cost, if possible, with appropraite reference) would be beneficial.
  • Line 137-138 - do you mean targeted sorption of specific metal ions, rather than any?
  • The experimental section is comprehensive, though in Lines 227-233 there are numerous references to Europium, which is not used in this work. Do you mean Scandium and Neodymium? Was this just copied from previous manuscripts. Please change any such references where they occur, so as not to mislead the reader.
  • What scale were the sorption analyses performed on? Is this clearly stated in the manuscript as I can't find the values. Please state the mass (mg/g per litre) and volume (ml or L) of solution used.
  • The results and discussion are comprehensive with a good comparision in the charaterisation before and after sorption of metal ions.

There are a few points where the English is a little disjointed, so the manuscript may benefit from review by an English editing service, but beyond this, I am happy to accept this piece for publication following the addressing of the above points.

Author Response

Dear Reviewer!

Thank you very much for your notes. Answers to it are presented below in a numerical order.

Note 1:

Line 38 - Scandium is a very rare element and has very few niche uses, outside of alloys in limited applications, given worldwide produciton is < 20 tons/year. Nd is far more useful and processed on a far larger scale. Please correct this and in lines 47-64. This paragraph can likely be condensed and merged with the one discussing Nd applicaitons. Statement of total worldwide production (using latest values of mass, and cost, if possible, with appropraite reference) would be beneficial.

Answer:

According to this note, the corrected version of the manuscript contains modified information about Scandium. As was advised the two single paragraphs (about neodymium and scandium) were merged into one.

Note 2:

Line 137-138 - do you mean targeted sorption of specific metal ions, rather than any?

Answer:

It was meant that using of remote interaction of polymer structures provides possibility of development of selective sorbents (or sorption systems) for selective sorption of aimed metal ion. The mentioned Lines were rewritten for more clear understanding.

Note 3:

The experimental section is comprehensive, though in Lines 227-233 there are numerous references to Europium, which is not used in this work. Do you mean Scandium and Neodymium? Was this just copied from previous manuscripts. Please change any such references where they occur, so as not to mislead the reader.

Answer:

Our previous work was devoted to sorption of Europium ions, the experimental section was just copied from the previous paper. Surely it was meant that these methods are used for determination of Scandium and Neodymium ions. The corresponding changes are made in the experimental section. Also it should be noted to avoid plagiarism the section was paraphrased.

Note 4:

What scale were the sorption analyses performed on? Is this clearly stated in the manuscript as I can't find the values. Please state the mass (mg/g per litre) and volume (ml or L) of solution used.

Answer:

Please pay your attention to lines 192-194. According to information presented in these lines two salts (Neodymium sulfate hydrate and Scandium sulfate hydrate (concentration of both is 100 mg/L) were used as model salts solution for the sorption experiments. Solutions were prepared with application of deionized water (χ=10 µS/cm; pH=6.98).

Note 5:

The results and discussion are comprehensive with a good comparision in the charaterisation before and after sorption of metal ions.

Answer:

Thank You very much. We did our best for comprehensive study of sorption properties (extraction degree, polymer chain binding degree, effective dynamic exchange capacity) of the interpolymer system based on Amberlite IR120 and AB-17-8 ion-exchangers during sorption of neodymium and scandium ions.

Also I would like to note that somewhere the text of the original manuscript undergo moderate changes.

Reviewer 2 Report

If the adsorption is based on ion exchange, it should reach equilibrium in a short time. Why does the experiment need several hours or even tens of hours?

Are there any similar work reports? How does your work compare with your existing work?

Has the adsorption thermodynamics of the above related ions been studied?

Author Response

Dear Reviewer!

Thank you very much for your notes. Answers to it are presented below in a numerical order.

Note 1:

If the adsorption is based on ion exchange, it should reach equilibrium in a short time. Why does the experiment need several hours or even tens of hours?

Answer:

Due to the fact that remote interaction phenomenon of Amberlite IR120 and AB-17-8 in the interpolymer system on their basis leads to their transfer into highly ionized state the sorption process need more than 24 hours for reaching the maximum sorption of neodymium and scandium ions.

Note 2:

Are there any similar work reports? How does your work compare with your existing work?

Answer:

Our previous work Effective Sorption of Europium Ions by Interpolymer System Based on Industrial Ion-Exchanger Resins Amberlite IR120 and AB-17-8 (https://doi.org/10.3390/ma14143837) was devoted to study of europium ions sorption features by the interpolymer system Amberlite IR120 – AB-17-8. In the previous work it was found that remote interaction effect of the ion-exchangers leads to significant increase of the initial values of the sorption properties.

The present work’s aim is to study the influence of the ionic radii of neodymium and scandium ions on the sorption process when using the interpolymer system Amberlite IR120 – AB-17-8 as a selective sorbent. The results and discussion are comprehensive with a comparision in the charaterisation before and after sorption of the rare-earth metal ions.

Note 3:

Has the adsorption thermodynamics of the above related ions been studied?

Answer:

Thank you very much for your note, really the impact of effect of temperature on rate of sorption is very important – predicted that increase of temperature will provide growth of sorption rate. These studies will be fully conducted in future works of our research group.